# Microbubbles and UltraSound-accelerated Thrombolysis (MUST) for peripheral arterial occlusions: protocol for a phase II single-arm trial

Harm P Ebben,[1,2] Johanna H Nederhoed,[1] Rutger J Lely,[3] Willem Wisselink,[1] Kakkhee Yeung,[1,2] on behalf of the MUST collaborators

► Prepublication history and additional material are available. To view these files please visit the journal online (http://dx.doi.org/ 10.1136/bmjopen-2016-014365).

[1]Departments of Surgery, VU University Medical Center, Amsterdam, the Netherlands
[2]Departments of Physiology, VU University Medical Center, Amsterdam, the Netherlands
[3]Departments of Radiology, VU University Medical Center, Amsterdam, the Netherlands

**Correspondence to**
Dr Kakkhee Yeung;
k.yeung@vumc.nl

## ABSTRACT

**Introduction** Acute peripheral arterial occlusions can be treated with intra-arterial catheter-directed thrombolysis as an alternative to surgical thromboembolectomy. Although less invasive, this treatment is time-consuming and carries a significant risk of haemorrhagic complications. Contrast-enhanced ultrasound using microbubbles could accelerate dissolution of thrombi by thrombolytic medications due to mechanical effects caused by oscillation; this could allow for lower dosages of thrombolytics and faster thrombolysis, thereby reducing the risk of haemorrhagic complications. In this study, the safety and practical applicability of this treatment will be investigated.

**Methods and analysis** A single-arm phase II trial will be performed in 20 patients with acute peripheral arterial occlusions eligible for thrombolytic treatment. Low-dose catheter-directed thrombolysis with urokinase will be used. The investigated treatment will be performed during the first hour of thrombolysis, consisting of intravenous infusion of 4 Luminity phials (6 mL in total, diluted with saline 0.9% to 40 mL total) of microbubbles with the use of local ultrasound at the site of occlusion. Primary end points are the incidence of complications and technical feasibility. Secondary end points are angiographic and clinical success, duration of thrombolytic infusion, treatment-related mortality, amputations, additional interventions and quality of life.

**Ethics and dissemination** Ethical approval for this study was obtained in 2015 from the Medical Ethics Committee of the VU University Medical Center, Amsterdam, the Netherlands. A statement of consent for this study was given by the Dutch national competent authority. Data will be presented at national and international conferences and published in a peer-reviewed journal.

**Trial registration numbers** Dutch National Trial Registry: NTR4731; European Clinical Trials Database of the European Medicines Agency: 2014-003469-10; Pre-results.

## Strengths and limitations of this study

► This will be a first in man study to examine the safety and technical feasibility of therapeutic microbubbles, combined with the application of ultrasound and catheter-directed thrombolysis in peripheral arterial occlusions.

► This is a 'single-arm' trial. The data will be used to inform a future large multicentre randomised controlled trial comparing conventional catheter-directed thrombolysis with Microbubble and UltraSound-accelerated Thrombolysis.

► The present study is a non-randomised phase II trial; therefore, the results cannot confirm benefit of sonothrombolysis for peripheral arterial occlusions, only safety and feasibility is analysed.

► The present study does not compare other thrombolysis techniques or protocols.

in amputation or death if not treated successfully.[1] Intra-arterial infusion of thrombolytic agents, that is, catheter-directed thrombolysis, can restore blood flow by dissolving the clot, as a less invasive alternative to surgical thromboembolectomy.[2] In comparison with the lysis of small arterial occlusions in patients with myocardial infarction, larger peripheral arterial occlusions require higher doses of lytic agents and infusion over a longer period of time. Inevitably, such treatment is accompanied by a risk of major haemorrhagic complications, such as haemorrhagic stroke, in up to 8% of patients.[3] Furthermore, this technique is time-consuming (several days of bed rest is usually required) and repeated angiography for treatment monitoring is needed, putting patients at risk for contrast-induced nephropathy. As a result, this leads to high morbidity rates and a significant patient burden. Methods to improve this therapy are therefore highly sought after.

A potential accelerator of thrombolytic therapy is contrast-enhanced ultrasound.

## INTRODUCTION

Acute limb ischaemia can be caused by a thrombus occluding an artery in an arm or leg. This is an emergency situation that can result

Contrast agents, consisting of 5–10 µm gas-filled particles (microbubbles), have initially been used as diagnostic ultrasound contrast enhancers. A new field of research investigates these agents for potential therapeutic purposes such as targeted drug delivery and thrombolysis.[4] The proposed mechanism of action in thrombolysis is that microbubbles can oscillate under the influence of ultrasound. At high intensities, this oscillation can lead to microbubble collapse and the production of mechanical forces on the clot surface, making the thrombus more susceptible to thrombolytics, thus accelerating thrombolysis.[5]

In early stages of clinical research, this technique has been shown to be efficient as treatment for acute cerebral stroke and acute myocardial infarction.[6 7] Although the safety of their clinical administration in treating smaller arteries in the heart has been a topic of discussion in the past, postmarketing data for diagnostic indications showed continued safety after extensive research in more recent years.[8–10] For therapeutic thrombolytic purposes, this technique has been shown to be effective and safe in a porcine model of large peripheral arterial occlusions.[11] In this study, we will investigate the therapeutic application of microbubbles with ultrasound in combination with catheter-directed thrombolysis for patients with peripheral arterial occlusions. An illustrative video regarding our research project is available as online supplementary video.

## METHODS AND ANALYSIS
### Study objectives
To investigate the safety and practical applicability of the therapeutic application of microbubbles and ultrasound in combination with catheter-directed thrombolysis for patients with peripheral arterial occlusions.

### Design
The Microbubbles and UltraSound-accelerated Thrombolysis (MUST) trial is a single-arm phase II trial.

### Primary study parameters
Main end points will be the safety and technical feasibility of the experimental treatment. Safety will be determined by treatment-related mortality, the occurrence of adverse events (AEs), serious adverse events (SAEs) and suspected unexpected serious adverse reactions (SUSARs). AEs will be defined as any undesirable experience occurring to a subject during the experimental treatment period, whether or not considered to be related to the investigational drug or intervention. SAEs will be defined as any untoward medical occurrence or effect that at any dose results in death; is life threatening (at the time of the event); requires hospitalisation or prolongation of existing in-patients' hospitalisation; results in persistent or significant disability or incapacity; is a new event of the trial likely to affect the safety of the subjects, such as an unexpected outcome of an adverse reaction. SUSARs, which are related to the microbubble infusion and ultrasound application, are the formation of microemboli resulting in occlusion of the microcirculation, haemorrhages, hypotension, heart rhythm disorders and anaphylaxis. See the section 'Adverse events' for detailed handling procedures of AEs, SAEs and SUSARs. Haemorrhagic complications related to thrombolytic therapy will be reported according to the Standardized Bleeding Definitions for Cardiovascular Clinical Trials proposed by Mehran et al.[12]

Technical feasibility will be defined as accomplishment of the experimental protocol during the first hour of thrombolysis.

### Secondary study parameters
Angiographic success will be defined as dissolution of >95% of the thrombus with outflow to at least one crural artery. Clinical change/success will be reported according to Rutherford's recommended scale for gauging changes in clinical status (table 1). Amputations will be defined as either major (above or below knee amputation) or minor (metatarsal or toe amputation).

| Table 1 | Rutherford's[24] recommended scale for gauging changes in clinical status |
| --- | --- |
| +3 | Markedly improved: no ischaemic symptoms and any foot lesions completely healed; ABI essentially 'normalised' (increased to >0.90) |
| +2 | Moderately improved: no open foot lesions; still symptomatic but only with exercise and improved by at least one clinical chronic ischaemia category; ABI not normalised but increased by >0.10 |
| +1 | Minimally improved: >0.10 increase in ABI* but no categorical improvement or vice versa (ie, upward categorical shift without an increase in ABI of >0.10) |
| 0 | No change: no categorical shift and <0.10 change in ABI |
| −1 | Mildly worse: no categorical shift but ABI decreased >0.10 or downward categorical shift with ABI decrease <0.10 |
| −2 | Moderately worse: one category worse or unexpected minor amputation |
| −3 | Markedly worse: more than one category worse or unexpected major amputations |

*In cases where the ABI cannot be accurately measured, an index based on the toe pressure, or any measurable pressure distal to the site of revascularisation, may be substituted.
ABI, Ankle Brachial Index.
Adapted from Rutherford RB et al. JVS 1997.

Additional interventions will be categorised as either surgical (eg, thromboembolectomy, bypass graft surgery) or percutaneous (percutaneous transluminal angioplasty, stent placement) and as either required for restoration of patency or necessary for correction of underlying lesions. We will also determine microcirculation of the limb (by Laser Doppler measurements, Perimed Instruments, Järfälla, Sweden), 30-day mortality, conversion to surgery, serum fibrinogen concentrations measured during thrombolytic treatment on a daily basis, pain by visual analogue scale (VAS) and quality of life by SF-36 questionnaires. The duration of thrombolysis will be defined by the time span between initiation and completion angiography.

### Patients and eligibility criteria

The present feasibility and safety study is a non-randomised phase II trial to be conducted in our university hospital in Amsterdam, the Netherlands. Usually, in a phase II trial 10–20 patients are investigated to confirm an occurrence of toxic effects or SAEs<20%. We chose a sample size of 20 to assess the safety of the investigational treatment. Eligibility criteria are listed in table 2. Inclusion of 20 eligible patient is expected within 1.5 years. Written informed consent will be acquired by a member of the Research Team after information about the study has been provided by the treating doctor.

### Data handling

We will keep an electronic log of patients who fulfil the eligibility criteria, patients who are invited to participate in the study, patients recruited and patients who withdraw from the study. Reasons for non-recruitment will also be recorded. We will attempt to collect reasons for non-participation from patients who decline to take part. During the course of the study, we will document reasons for withdrawal from the study and loss to follow-up. Data will be stored electronically in Case Report Forms software with audit trail functionality and will be audited by the institutional Clinical Research Bureau. Only

| Table 2 | Eligibility criteria |
|---|---|
| Inclusion criteria | ► Men and women older than 18 and younger than 85 years<br>► Patients with a maximum of 2 weeks of symptoms for lower limb ischaemia due to thrombosed/occluded iliofemoral, femoropopliteal or femorocrural native arteries or iliofemoral, femoropopliteal or femorocrural venous or prosthetic bypass grafts<br>► Patients appropriate for thrombolysis, that is, with acute lower limb ischaemia class I and IIa according to the Rutherford classification<br>► Patients who understand the nature of the procedure and provide written informed consent before enrolment in the study |
| Exclusion criteria | ► Patients with clinical complaints of acute lower limb ischaemia due to thrombosis of iliofemoral, femoropopliteal or femorocrural native arteries, or iliofemoral, femoropopliteal or femorocrural venous or prosthetic bypass grafts for >2 weeks<br>► Patients with thrombosed popliteal aneurysms<br>► Patients with absolute contraindications for administration of antiplatelet therapy, anticoagulants or thrombolytics<br>► History of recent (<6 weeks) ischaemic stroke, cerebral haemorrhagic or myocardial infarction<br>► Patients with recent (<6 weeks) surgery<br>► Severe hypertension (diastolic blood pressure >110 mm Hg and/or systolic blood pressure >200 mm Hg)<br>► Current malignancy or severe comorbid condition with a life expectancy of <6 months<br>► Patients with uncorrected bleeding disorders (gastrointestinal ulcer, menorrhagia, liver failure)<br>► Women with childbearing potential not taking adequate contraceptives or currently breast feeding<br>► Pregnancy<br>► Patients who are currently participating in another investigational drug or device study<br>► Patients younger than 18 years or older than 85 years<br>► Patients with contraindications for Luminity microbubbles, that is:<br> ► Hypersensitivity to perflutren or to any of the components of Luminity<br> ► Recent acute coronary syndrome or clinically unstable ischaemic cardiac disease, including evolving or ongoing myocardial infarction, unstable angina at rest within the last seven days, significant worsening of cardiac symptoms within the last seven days, recent coronary artery intervention or other factors suggesting clinical instability (eg, recent deterioration of ECG, laboratory or clinical findings), acute cardiac failure, class III/IV cardiac failure or severe rhythm disorders<br> ► Patients known to have right-to-left cardiac shunts, severe pulmonary hypertension (pulmonary artery pressure>90 mm Hg), uncontrolled systemic hypertension and in patients with Global Initiative for Obstructive Lung Disease (GOLD) stage IV chronic obstructive pulmonary disease, diffuse interstitial fibrosis or adult respiratory distress syndrome<br> ► Patients with cardiovascular instability where dobutamine is contraindicated |

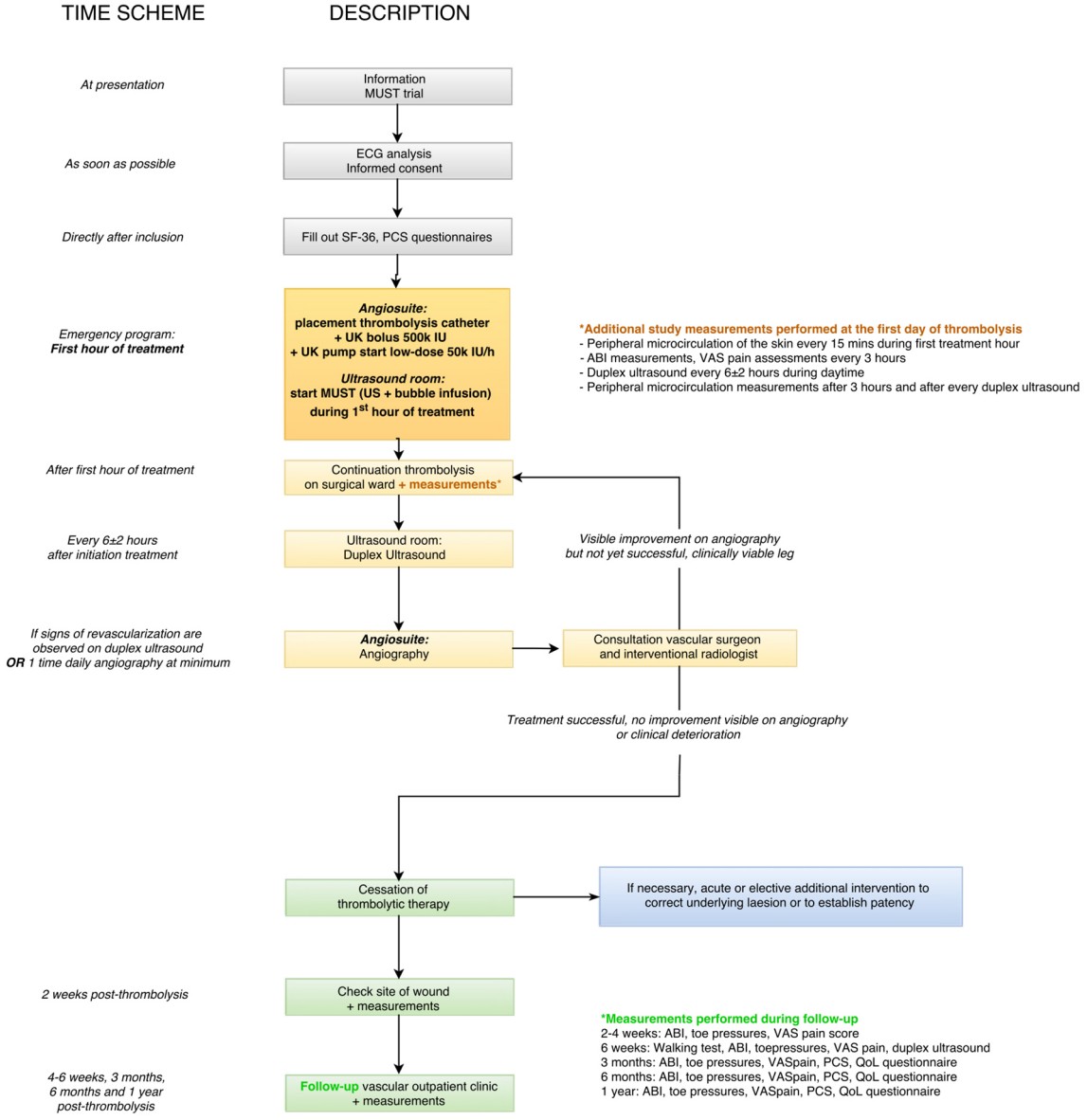

TIME SCHEME          DESCRIPTION

*At presentation* — Information MUST trial

*As soon as possible* — ECG analysis Informed consent

*Directly after inclusion* — Fill out SF-36, PCS questionnaires

*Emergency program:* **First hour of treatment** — **Angiosuite:** placement thrombolysis catheter + UK bolus 500k IU + UK pump start low-dose 50k IU/h **Ultrasound room:** start MUST (US + bubble infusion) during 1st hour of treatment

*Additional study measurements performed at the first day of thrombolysis
- Peripheral microcirculation of the skin every 15 mins during first treatment hour
- ABI measurements, VAS pain assessments every 3 hours
- Duplex ultrasound every 6±2 hours during daytime
- Peripheral microcirculation measurements after 3 hours and after every duplex ultrasound

*After first hour of treatment* — Continuation thrombolysis on surgical ward + measurements*

*Every 6±2 hours after initiation treatment* — Ultrasound room: Duplex Ultrasound

*Visible improvement on angiography but not yet successful, clinically viable leg*

*If signs of revascularization are observed on duplex ultrasound OR 1 time daily angiography at minimum* — **Angiosuite:** Angiography — Consultation vascular surgeon and interventional radiologist

*Treatment successful, no improvement visible on angiography or clinical deterioration*

Cessation of thrombolytic therapy — If necessary, acute or elective additional intervention to correct underlying laesion or to establish patency

*2 weeks post-thrombolysis* — Check site of wound + measurements

*Measurements performed during follow-up
2-4 weeks: ABI, toe pressures, VAS pain score
6 weeks: Walking test, ABI, toepressures, VAS pain, duplex ultrasound
3 months: ABI, toe pressures, VASpain, PCS, QoL questionnaire
6 months: ABI, toe pressures, VASpain, PCS, QoL questionnaire
1 year: ABI, toe pressures, VASpain, PCS, QoL questionnaire

*4-6 weeks, 3 months, 6 months and 1 year post-thrombolysis* — **Follow-up** vascular outpatient clinic + measurements

**Figure 1** Flow chart of patient work-up after presentation. ABI, Ankle Brachial Index; IU, International Units; MUST, Microbubbles and UltraSound-accelerated Thrombolysis; PCS, Pain Catastrophising Scale; QoL, quality of life; UK, urokinase; US, ultrasound; VAS, visual analogue scale.

anonymised information will be stored and participants will only be identifiable by their unique study number, which will be kept in a separate file. Data will be securely stored on these servers for 15 years according to national guidelines. The principal investigator will have access to the final trial data set. No independent Data Management Committee was instated according to local ethics committee guidelines since the present study was not classified as a high-risk clinical study. This classification was approved by the local ethics committee based on the risk assessment form of the Netherlands Federation of University Medical Centres.

## Study procedures
### Intervention
A flow chart of the patient work-up after presentation in our hospital is presented in figure 1. Low-dose

thrombolytic treatment with urokinase (UK) will be initiated following our standard institutional protocol: a catheter is placed intra-arterially in the affected artery and a bolus injection of 500 000 International Units (IU) of UK (Medacinase Urokinase, Medac, Hamburg, Germany) will be followed by the continuous infusion of 50 000 IU of UK per hour and 9600 IU of heparin per 24 hours. The experimental treatment consists of (in addition to the standard thrombolytic therapy) the use of local 1.8 MHz transdermal ultrasound (Philips iE33 Ultrasound Machine, Eindhoven, the Netherlands), and the intravenous infusion of 4 Luminity phials (total 6 mL, diluted with saline 0.9% to 40 mL total, Lantheus MI UK, Newbury, Berkshire, UK) during the first hour of thrombolysis with U. An ACIST VueJect (Bracco Imaging Europe B.V., the Netherlands) infusion pump will be

used to infuse the four phials continuously. Ultrasound will be intermittently activated (3 s manual flash to burst microbubbles with Mechanical Index (MI) 1.08 (pulse duration 20 µs, frequency 1.8 MHz, frame rate 39 Hz), 7 s of visualisation of inflow of the microbubbles at MI±0.11), at the site of occlusion during the first hour of thrombolysis. Criteria for discontinuation of the experimental treatment during the first hour will be the occurrence of any AEs potentially related to the experimental treatment, such as bleeding and allergic reactions.

## Assessments

### Diagnostic measurements
Additional diagnostic measurements during admission including ECG, duplex ultrasound, angiography and microcirculation measurements (by Laser Doppler flowmetry) will be performed as depicted in figure 1.

A duplex ultrasound will be performed every 6±2 hours to monitor for signs of revascularisation. When resumption of flow is visualised by duplex ultrasound, angiography will be performed to confirm flow. Angiography will be performed at least once daily as standard procedure. Outside of routine hospital working hours, angiography will only be performed in emergencies as per standard care.

A standardised pain score (VAS, 1–10) and Pain Catastrophising Scale will be recorded every 3 hours by a nurse practitioner, research fellow or surgical resident to assess pain.

### Fibrinogen monitoring
Following our standard institutional thrombolysis protocol, fibrinogen concentration will be checked during thrombolysis with the following criteria for treatment modification: if <1.0 g/L, the UK infusion rate will be lowered to 25 000 IU/hour; if <0.5 g/L, thrombolysis must be aborted temporarily and replaced by normal saline infusion. Three hours following treatment modifications, fibrinogen concentration will be re-evaluated and when >1.0 g/L thrombolysis will be restarted at an initial low-dose urokinase of 50 000 IU/hour.

### Postprocedural anticoagulation
After successful thrombolysis, the patient will be heparinised with low-molecular-weight heparin (fraxiparine) dosed based on body weight:<50 kg: two times a day 3800 IU (= 0.4 mL), 50–80 kg: two times a day 5700 IU (= 0.6 mL), >80 kg: two times a day 7600 IU (= 0.8 mL).

Concomitant therapy with coumarin derivatives will also be started at that time. Activated partial thromboplastin time will be measured daily during heparin treatment. The target range international normalised ratio will be 2.5–3.5; if this value is reached, heparinisation will be stopped and coumarin treatment will be continued.

### Follow-up
Outpatient follow-up will take place at specific time points for a total duration of 1 year; measurements performed during follow-up visits are depicted in figure 1.

## Adverse events
AEs will be recorded in detail in the electronic patient record. Any SAEs that occur after joining the trial will be reported to the accredited Medical Ethics Committee of our institution according to national and institutional guidelines. All AEs will be followed up until they have abated or until a stable situation has been reached. Depending on the event, follow-up may require additional tests or medical procedures as indicated and/or referral to a general physician or medical specialist. An interim analysis after 10 patients will be performed. If SAEs in these 10 patients have occurred, we will discuss the continuation of the study. The study will be prematurely terminated if two or more intracranial bleedings occur or more than five allergic reactions.

## Statistical analysis
Categorised epidemiological/descriptive patient variables are summarised with frequencies and will be analysed with Fischer's exact test or Pearson's $\chi^2$ test. To avoid possible violations of the assumptions for parametric testing, such as a normal distribution pattern, we will employ non-parametric methods such as Spearman's rank correlation and Mann-Whitney U test in the case of a skewed distribution or log transformation. For associations of two outcome measurements, we will use a correlation (Spearman's rank) or single regression analysis. We will analyse the following outcomes by means of Kaplan-Meier curves: patency rate, amputation-free rate and intervention-free rate. We will assess heterogeneity in prognostic factors as a secondary analysis by means of $\chi^2$ tests. All tests will be performed two-sided, and $p < 0.05$ will be considered to be statistically significant.

## ETHICS AND DISSEMINATION
The study will be conducted according to the principles of the Declaration of Helsinki (Brazil, October 2013), and in accordance with the Medical Research Involving Human Subjects Act . An Investigator Site File will be produced in advance of the study conforming to institutional guidelines. Furthermore, we will create Case Report Forms by using Good Clinical Practice and 21 Code of Federal Regulations Part 11 compliant software to handle patient data.

The study has been registered in the Dutch Trial Register, at the Dutch National Central Committee on Research Involving Human Subjects (CCMO) and in the European Clinical Trials Database of the European Medicines Agency. Any protocol amendments during the study will be communicated and changed accordingly in the relevant registries after approval of the institutional Medical Ethics Committee. The results of this study will be submitted for publication in a peer-reviewed journal, regardless of the outcome of this study, according to the CCMO statement on publication policy. Data will also be presented at national and international conferences.

## DISCUSSION

The MUST trial is a phase II single-arm clinical trial. In this study, the safety and feasibility of an experimental ultrasound technique will be investigated for the first time in patients with large peripheral arterial occlusions.

We believe that this procedure is safe and can accelerate thrombolysis, thereby allowing for reduction of thrombolytic dosage, which in turn reduces the risk of major haemorrhagic complications.

An experimental bolus therapy with microbubbles and ultrasound could accelerate thrombolysis because at high ultrasound intensities microbubbles can collapse, resulting in mechanical forces on the clot surface. The formation of small channels in the thrombus leads to exposure of a larger total surface susceptible to thrombolytics.[5]

In regard to the therapeutic application of contrast agents, several studies have been performed in patients with ischaemic stroke and myocardial infarction. A systematic review of sonothrombolysis shows that this treatment option improves short-term and long-term clinical outcomes, while potentially reducing bleeding risk, in patients with ischaemic stroke.[13] Nevertheless, dose escalation studies show that the safety (in terms of bleeding and microemboli) needs to be further investigated before enrolling patients in phase III trials.[14] Few and heterogeneous studies examined the therapeutic application of sonothrombolysis in patients with myocardial infarction. Although pilot studies affirm safety and feasibility, the application of therapeutic ultrasound with longer pulse durations (20 vs 5 μs) was reported to result in unexpected coronary vasoconstriction in a recent clinical trial.[15]

Potential reported mechanisms for this effect are the summative effect of myocardial ischaemia, reperfusion damage and long-pulse-duration sonoporation on endothelial damage, all leading to calcium overload.

However, patients with peripheral arterial occlusions are mostly chronic vascular patients who often have received previous treatments in the respective artery, for example, thrombolytic therapy, percutaneous transluminal angioplasty, thromboembolectomy or bypass surgery. The mechanical manipulation of the vascular wall during all these treatments is extensive. Furthermore, during standard thrombolytic treatment, arteries are manipulated and perforated on purpose to insert guide wires and catheters. Hence, vascular spasms during these peripheral treatments are normal and non-threatening to the patient in contrast to spasms in small coronary arteries.

The administration of ultrasound contrast agents has been accompanied by important discussions regarding safety concerns in the past.[8 16] As a response to the occurrence of SAEs, the US Food and Drug Administration issued a labelling change and warnings for contrast agents in 2007. Consequently, new studies on the risks of contrast agents were performed and these established their safety.[17]

In regard to Luminity contrast agent dose regimens, the recommended dose for diagnostic indications is 1.3 mL dispersion added to 50 mL of sodium chloride 9 mg/mL (0.9%) or glucose 50 mg/mL (5%) solution injected over a short time period.[18] For therapeutic purposes, in large peripheral arteries there are no dose studies available. However, in our university hospital the Sonolysis study has been performed by our Cardiology department to treat acute ST elevation myocardial infarction patients with Luminity microbubbles and high mechanical ultrasound.[19] The dose used was one flacon Luminity of 1.5 mL which contains 225 μL perflutren diluted with 48.5 mL of saline 0.9% to create a 50 mL suspension. Patients were treated for 15 min with an infusion rate of 200 mL/h. No SAEs occurred. In the present study to establish a therapeutic effect in large arterial occlusion, we will also infuse one phial per 15 min but we will treat patients for 60 min. We will use four flacons of 1.5 mL Luminity containing 900 μL perflutren diluted with saline 0.9% to 40 mL total volume to be infused during 1 hour. The clinical consequences of overdose with Luminity are not known. Single doses of up to 100 μL dispersion/kg and multiple doses up to 150 μL dispersion/kg were tolerated well in phase I clinical trials.[20] This equals to the infusion of 12 mL (eight flacons) of Luminity dispersion. We will administer a total of 6 mL (four flacons) of Luminity dispersion. Furthermore, we will not administer them as single bolus doses but as low-speed continuous infusion. During the experimental protocol with microbubble infusion, patients will be continuously monitored.

As with all contrast agents, the risk of anaphylactic reactions to contrast remains. Therefore, administration of contrast agents in a centre with full resuscitation possibilities is mandatory. Furthermore, during the first hours of administration, monitoring of vital parameters of patients is important.

In this study, thrombolysis is performed with the fibrinolytic urokinase, which is the most used fibrinolytic agent for the treatment of peripheral arterial occlusions worldwide and is standard care in the Netherlands. Some countries use tissue plasminogen activator for this indication. A Cochrane review on the topic states that there is no evidence that (r)t-PA is more effective than urokinase for patients with peripheral arterial occlusion.[21]

If the application of microbubbles and ultrasound concomitant to catheter-directed thrombolysis is shown to be safe and technically feasible based on this phase II trial, we anticipate a funding application for a larger randomised controlled trial with a comparative group to assess and compare efficacy of this treatment.

Although the efficacy of the currently described protocol cannot be adequately compared within this study design, we will discuss the outcomes relative to a historic control group that had previously received our standard hospital thrombolysis protocol.[22]

Successful thrombolysis is strongly predictive of amputation-free survival with vascular patency for at least 1 year.[23] A longer duration of thrombolysis inevitably exposes a

patient to a higher thrombolytic dose and higher risk of haemorrhage, in addition to an already increased patient burden because of prolonged bed rest. Therefore ultimately, acceleration of thrombolysis with microbubbles could benefit the patient because of a shorter therapy time, a lower risk of haemorrhagic complications and a decrease in patient burden.

**Acknowledgements**  The authors thank Laura van Wieringen, Lloyd Belliot, Joyce Lu, Jorn Meekel, Ted van Schaik and Sean Matheiken for their assistance in preparation of this study.

**Collaborators**  Arjan WJ Hoksbergen and Jan D Blankensteijn, Department of Surgery, VU University Medical Center, Amsterdam, The Netherlands; Bram B van der Meijs and Martijn R Meijerink, Department of Radiology, VU University Medical Center, Amsterdam, The Netherlands; Otto Kamp, J Slikkerveer, Department of Cardiology VU University Medical Center, Amsterdam, The Netherlands; René JP Musters and Geert-Jan Tangelder, Department of Physiology, VU University Medical Center, Amsterdam, The Netherlands.

**Contributors**  HE, JN, WW and KK planned and designed the research. HE wrote the manuscript. HE, JN, RL and KK critically revised the manuscript and all authors and MUST collaborators approved the final version of the manuscript.

**Funding**  This study is supported by the Departments of Surgery, Radiology, Cardiology and Physiology of the VU University Medical Center Amsterdam, the Netherlands and by a unrestricted research grant from LamePro B.V., the Netherlands. LamePro B.V. has no role in the study design, collection, management, analysis, and interpretation of data, nor in writing of the report, submitting the report for publication or ultimate authority on any of the previous mentioned.

**Competing interests**  None declared.

**Ethics approval**  Medical Ethical Committee VU University Medical Center.

**Provenance and peer review**  Not commissioned; externally peer reviewed.

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
