## [Reviewer comments · BMJ Open]

ARTICLE DETAILS

TITLE (PROVISIONAL)	Microbubbles and UltraSound-accelerated Thrombolysis (MUST) for peripheral arterial occlusions: protocol for a phase-II single arm trial
AUTHORS	Ebben, Harm; Nederhoed, Johanna; Lely, Rutger; Wisselink, Willem; Yeung, Kakkhee; MUST, Collaborators

VERSION 1 - REVIEW

REVIEWER	P Bagan CH V Dupouy, France
REVIEW RETURNED	05-Oct-2016

GENERAL COMMENTS	The authors should provide more details, explanation and justification concerning the post procedural anticoagulation protocol.
---

REVIEWER	Christian H Nolte Department of Neurology Charite-Universitätsmedizin Berlin Germany
REVIEW RETURNED	10-Oct-2016

GENERAL COMMENTS	This is the protocol of a pilot study on Microbubbles and Ultrasound-accelerated thrombolysis in peripheral artery occlusion. This is a very interesting subject. However, this study does not compare different therapeutic approaches. This is a pilot study planned to collect data on complications (safety), technical feasibility, mortality and other secondary endpoints. The planned sample size is N=20. - Please justify the planned sample size. Will a sample size of N=20 allow for reliable estimates to plan for a proper phase 2 trial? What complications rates are likely to be estimated (below 20%? below 10%?....)- Please define Ultrasound application ("the use of local ultrasound": What frequency will be applied? Later you state 1.8 MHz. Please state frequency earlier in the text). Frequency is very relevant.- Please define how you will measure "dissolution of >95% of the thrombus" (secondary study parameter). What is the method that will obtain information on dissolution?- Please give a reason why patients of 85years and older are not included (why is there an upper age limit)- Please define "appropriateness" for thrombolysis (inclusion criterion). Which criteria will be applied?- Please discuss the drug. Why do you use Urokinase and not rTPA?
--

	 - Please define time duration of ultrasound application - Please define time points of Miccorbubble application ("4 vials during the first hour of thrombolysis"- ?Intervals of 10 Minutes? Every 20 Minutes? Differen intervals? - Page 9 of 19, section "diagnostic measurements": word missing in first sentence: "measurements during....including ECG" - Page 9 of 19: Statistical analysis: What cut-off was considered relevant for skewness? ("in the case of skewed distribution") - Discussion (page 11 of 19). Define "safe". You will have a complication rate. What complicatins rate will be considered safe? Below 5%? Below 10%? What complication rate maybe estimated with N=20? - Page 12 of 19: If you plan to compare this pilot to a historic control group. Did this help to calculate the sample size? Do you hypothesize that this approach will be inferior, superior or non-inferior if compared to the historic cohort? How did you calculate the sample size? (Alpha, Power....)
--	--

REVIEWER	Dr. Karin Pfister University Hospital Regensburg, Germany Department of Vascular Surgery Medac Support for a study to evaluate safety in intraoperative use of Urokinase in acute limb ischemia
REVIEW RETURNED	13-Dec-2016

GENERAL COMMENTS	Excellent idea to evaluate a protocol in patients with peripheral arterial occlusions comparable to former cerebrovascular studies using sonothrombolysis Nevertheless there are important technical aspects that need to be addressed:  1. What data/research are the basis of the protocol according to medication (urokinase or rt-PA) dosage scheme, application rate of heparin or sonovue or ultrasound parameters (frequency, frame rate, duration)? 2. pls include or specify the length of occlusion in the inclusion criteria. In summary I think, it was a good idea to create this protocol, so I recommend publication with minor revisions
---

VERSION 1 – AUTHOR RESPONSE

Reviewer: 1
Reviewer Name: P Bagan
Institution and Country: CH V Dupouy, France
Competing Interests: None declared

The authors should provide more details, explanation and justification concerning the post procedural anticoagulation protocol.

Details on our post-procedural anticoagulation protocol are described in the paragraph 'post-procedural anticoagulation' in the methods section of page 9 of our manuscript and is based on our institutional protocol. Our institutional protocol is in accordance with national guidelines and also

international guidelines. (for reference: Quality Improvement Guidelines for Percutaneous Catheter-Directed Intra-Arterial Thrombolysis and Mechanical Thrombectomy for Acute Lower-Limb Ischemia. *Cardiovasc Interv Radiol*. 2011;1123–36.)

Reviewer: 2

Reviewer Name: Christian H Nolte

Institution and Country: Department of Neurology, Charite-Universitätsmedizin Berlin, Germany

Competing Interests: None declared

This is the protocol of a pilot study on Microbubbles and Ultrasound-accelerated thrombolysis in peripheral artery occlusion. This is a very interesting subject.

However, this study does not compare different therapeutic approaches. This is a pilot study planned to collect data on complications (safety), technical feasibility, mortality and other secondary endpoints. The planned sample size is N=20.

- Please justify the planned sample size. Will a sample size of N=20 allow for reliable estimates to plan for a proper phase 2 trial? What complications rates are likely to be estimated (below 20%? below 10%?....)

Phase-II clinical feasibility trials are designed to provide more detailed information about the safety of the treatment, in addition to evaluating if the technique is feasible in a clinical situation. The sample size of 20 is not powered to demonstrate statistically significant differences in lytic efficacy nor in bleeding complications. Bleeding complication rates below 10% are estimated since we use a low-dose thrombolysis protocol.

- Please define Ultrasound application ("the use of local ultrasound": What frequency will be applied? Later you state 1.8 MHz. Please state frequency earlier in the text). Frequency is very relevant.

We have appended the sentence on page 7 under the paragraph 'Study procedures' and subheading 'Intervention' to state 'the use of local 1.8 Mhz transdermal ultrasound', accordingly to the reviewer's suggestion.

- Please define how you will measure "dissolution of >95% of the thrombus" (secondary study parameter). What is the method that will obtain information on dissolution?

The measurement that will obtain information about thrombus dissolution is daily angiography. The secondary study parameter 'dissolution of >95% of the thrombus' is chosen in this study since the angiography method, despite being the golden standard, has not sufficient resolution to be able to assess 100% or complete dissolution of thrombus. The choice of stating the slight arbitrary '>95% dissolution' is based on this feature and furthermore used as a parameter in a recent clinical trial reporting angiographic success with this definition

(for reference: Schrijver et al. Dutch Randomized Trial Comparing Standard Catheter-Directed Thrombolysis and Ultrasound-Accelerated Thrombolysis for Arterial Thromboembolic Infrainguinal Disease (DUET). *J Endovasc Ther* [Internet]. 2015;22(1):87–95. Available from: <http://jet.sagepub.com/lookup/doi/10.1177/1526602814566578>).

- Please give a reason why patients of 85years and older are not included (why is there an upper age limit)

The rationale for including an upper age limit in our protocol is the relative contra-indication for thrombolytic therapy for patients of advanced age in our local thrombolysis protocol. This is based on the fact that

- Advanced age has been reported to be an independent risk factor for poor outcome and a higher risk of bleeding complications in acute myocardial infarction literature references (Gore JM et al (1995) Stroke after thrombolysis: mortality and functional outcomes in the GUSTO-I trial; Global Use of Strategies to Open Occluded Coronary Arteries. *Circulation* 92:2811–2818, Gurwitz JH et al (1998) Risk for intracranial hemorrhage after tissue plasminogen activator treatment for acute myocardial infarction: participants in the National Registry of Myocardial Infarction 2. *Ann Intern Med* 129:597–604).
- The instructions for use of the fibrinolytic agent urokinase used state special warnings and precautions for the use of urokinase in patients >75 years because of a higher risk of bleeding (stated in Dutch Summary of Product Characteristics but an equivalent is of this is described in the market authorisation document of the UK's Medicines and Healthcare products Regulatory Agency which can be found via <http://www.mhra.gov.uk/home/groups/par/documents/websiteresources/con108649.pdf>

However, the TOPAS and STILE trials in the 1990s and also more recent studies on thrombolytic therapy in patients with peripheral arterial occlusions have not reported high age to be a risk factor. (Kuoppala M et al. Risk factors for haemorrhage during local intra-arterial thrombolysis for lower limb ischaemia. *J Thromb Thrombolysis* [Internet]. Springer US; 2011 Feb 17;31(2):226–32)

Furthermore, exclusion of older people from clinical research is a controversial topic because of the aging population: the population over 80 years of age is growing and exclusion in clinical research would implicate improper evaluation of effects and outcomes in this age group.

Since our study mainly investigates safety and feasibility, we weighed the risks and benefits of incorporating an upper age limit in our protocol and concluded to an upper age limit of >85 years to not exclude all elderly but to exclude patients with a high risk for bleedings, also in concordance with the previously published DUET study on thrombolytic therapy.

- Please define "appropriateness" for thrombolysis (inclusion criterion). Which criteria will be applied?

We used, in accordance with our National Guideline 'Diagnosis and treatment of peripheral artery disease of the lower extremities', the following criteria: thrombolytic therapy is indicated in patients with non-marginally threatened limbs, i.e. patients with acute limb ischaemia class I and IIa according to the Rutherford classification of acute limb ischaemia (Vahl AC, Reekers JA. The guideline "Diagnosis and treatment of peripheral artery disease of the lower extremities" of The Netherlands Surgical Society. *Ned Tijdschr Geneesk.* Netherlands; 2005;149(30):1670–4., Rutherford RB, Baker JD, Ernst C, Johnston KW, Porter JM, Ahn S, et al. Recommended standards for reports dealing with lower extremity ischemia: revised version. *J Vasc Surg.* UNITED STATES; 1997;26(3):517–38.)

- Please discuss the drug. Why do you use Urokinase and not rTPA?

This is discussed in the discussion section on page 11 of our manuscript. We use the fibrinolytic Urokinase since Dutch vascular surgeons and interventional radiologists have the most clinical experience with this drug and it is almost exclusively used in our country for thrombolysis of peripheral arterial occlusions. We are aware of the use of other fibrinolytic agents such as (r)t-PA in some other countries including the UK and US, but a recent systematic review on fibrinolytic agents for this indication showed that there were no statistically significant differences between urokinase and (r)t-PA regarding efficacy and incidence of haemorrhagic complications.

- Please define time duration of ultrasound application

This is described in our manuscript under the paragraph 'Study procedures' and subheading 'Intervention' on page 7.

Ultrasound application will be performed during the first hour of thrombolytic treatment with concomitant continuous intravenous infusion of microbubbles. Ultrasound will be applied intermittently using a depletion-replenishment technique: ultrasound will be intermittently activated (3 seconds manual flash to burst microbubbles with Mechanical Index (MI) 1.08 (pulse duration 20 microseconds, frequency 1.8 Mhz, framerate 39 Hz), 7 seconds of visualization of inflow of the microbubbles at $MI \pm 0.11$, at the site of occlusion.

- Please define time points of Microbubble application ("4 vials during the first hour of thrombolysis" - ?Intervals of 10 Minutes? Every 20 Minutes? Different intervals?)

We described the timepoints and Microbubble application on page 7.

Microbubbles will be infused during the first hour of thrombolytic therapy by continuous infusion. By intermittent ultrasound application using a depletion-replenishment technique the microbubbles will be allowed to replenish into the thrombus during 7 seconds of low Mechanical Index (± 0.11) ultrasound application. The low MI ultrasound during these 7 seconds allows for visualization of the microbubbles flowing into the thrombus. High MI (± 1.1) ultrasound will be applied consecutively during 3 seconds to induce microbubble cavitation after which 7 seconds of microbubble replenishment is facilitated.

- Page 9 of 19, section "diagnostic measurements": word missing in first sentence: "measurements during....including ECG"

Thank you for this remark we have corrected the sentence in our manuscript with tracked changes on page 9 to clarify into 'Additional diagnostic measurements during admission including ECG, duplex ultrasound, angiography and microcirculation measurements (by Laser Doppler flowmetry) will be performed as depicted in Figure 1.'

- Page 9 of 19: Statistical analysis: What cut-off was considered relevant for skewness? ("in the case of skewed distribution")

We have mentioned 'skewed distribution' to implicate a non-parametric distribution. Data will be plotted to evaluate the distribution.

- Discussion (page 11 of 19). Define "safe". You will have a complication rate. What complication rate will be considered safe? Below 5%? Below 10%? What complication rate maybe estimated with $N=20$?

This phase II trial is designed to examine safety, which means analyzing serious adverse events related to our experimental therapy (i.e. sonothrombolysis with microbubbles). An interim analysis after 10 patients will be performed and if serious adverse events have occurred, we will discuss the continuation of the study. The study will be prematurely terminated if 2 or more intracranial bleedings occur or more than 5 allergic reactions.

We have added this detail to the manuscript on page 9.

Additionally, an important pitfall of thrombolytic therapy in general is the high risk of bleeding

complications in up to 13% of patients. In order to demonstrate a (statistically significant) decrease in bleeding complications between our patient group and rates reported in the literature a larger sample size would be required. However, this is not the scope of the present study. Theoretically, acceleration of thrombolysis could allow for lower dosages of fibrinolytics which in turn implies a lower risk of bleeding complications. The decrease in complication rate of this technique compared to the standard procedure would have to be demonstrated in a comparative study with larger sample size.

- Page 12 of 19: If you plan to compare this pilot to a historic control group. Did this help to calculate the sample size? Do you hypothesize that this approach will be inferior, superior or non-inferior if compared to the historic cohort? How did you calculate the sample size? (Alpha, Power....)

For this pilot study a sample size is chosen of 20 to evaluate the experimental treatment. The sample size is not calculated based on the historic control group because it is not a comparative study aimed to demonstrate differences between groups. However, the new treatment needs to work as well or better than the standard treatment for it to be further investigated in a phase-III clinical trial. We will use the historic control group for reference of our results to evaluate this.

Reviewer: 3

Reviewer Name: dr. Karin Pfister

Institution and Country: University Hospital Regensburg, Germany

Competing Interests: Medac Support for a study to evaluate safety in intraoperative use of Urokinase in acute limb ischemia

Excellent idea to evaluate a protocol in patients with peripheral arterial occlusions comparable to former cerebrovascular studies using sonothrombolysis

Nevertheless there are important technical aspects that need to be addressed:

1. What data/research are the basis of the protocol according to medication (urokinase or rt-PA) dosage scheme, application rate of heparin or sonovue or ultrasound parameters (frequency, frame rate, duration)?
2. pls include or specify the length of occlusion in the inclusion criteria.

In summary I think, it was a good idea to create this protocol, so I recommend publication with minor revisions

1. The thrombolysis regimen used in this protocol is the low-dose urokinase and heparin protocol administered in our university medical center as previously published (Reference no. 11 in our manuscript in the reference section on page 13) in Eur J Vasc Endovasc Surg. 2014 Aug; Available from: <http://www.sciencedirect.com/science/article/pii/S1078588414003888>

The ultrasound parameters used in the updated protocol are based on the experience in previous studies in acute coronary syndromes by our department of Cardiology

The dose regimen of the microbubbles is based on the available literature on the subject aiming at a maximal sonothrombolytic effect with a lower total dose that has previously been safely administered to patients and corrected for dose per weight. After acquiring approval of the Medical Ethics Committee to switch from microbubble brand SonoVue to Luminity we will provide the according details of the microbubble protocol.

2. The inclusion criteria is based on clinical findings, Rutherford classification and site of occlusion.

The precise length of occlusion is not specified. If catheter directed thrombolysis is possible and patients meet the inclusion criteria, we will include the patient in our study.